# Modeling Ozone Source Apportionment and Performing Sensitivity Analysis in Summer on the North China Plain

**Yujing Zhang** [1,2]**, Yuncheng Zhao** [1,2,*]**, Jie Li** [1,2,3,*]  **, Qizhong Wu** [4,5]**, Hui Wang** [4,5]**, Huiyun Du** [1,2]**, Wenyi Yang** [1,2,3]**, Zifa Wang** [1,2,3] **and Lili Zhu** [6]

1. State Key Laboratory of Atmospheric Boundary Layer Physics and Atmospheric Chemistry (LAPC), Institute of Atmospheric Physics, Chinese Academy of Sciences, Beijing 100029, China; zhangyujing@mail.iap.ac.cn (Y.Z.); duhuiyun@mail.iap.ac.cn (H.D.); yangwenyi@mail.iap.ac.cn (W.Y.); zifawang@mail.iap.ac.cn (Z.W.)
2. College of Earth and Planetary Sciences, University of Chinese Academy of Sciences, Beijing 100049, China
3. Center for Excellence in Regional Atmospheric Environment, Institute of Urban Environment, Chinese Academy of Sciences, Xiamen 361021, China
4. College of Global Change and Earth System Science, Beijing Normal University, Beijing 100875, China; wqizhong@bnu.edu.cn (Q.W.); huiw16@uci.edu (H.W.)
5. Joint Center for Global Changes Studies, Beijing Normal University, Beijing 100875, China
6. China National Environmental Monitoring Centre, Beijing 100012, China; zhull@cnemc.cn
* Correspondence: zhaoyuncheng@mail.iap.ac.cn (Y.Z.); lijie8074@mail.iap.ac.cn (J.L.)

**Abstract:** In recent years, air quality issues due to fine particulate matter have been sufficiently treated. However, ozone ($O_3$) has now become the primary pollutant in summer on the North China Plain (NCP). In this study, a three-dimensional chemical transport model (the Nested Air Quality Prediction Model System, NAQPMS) coupled with an online source apportionment module was applied to investigate the sources of $O_3$ pollution over the NCP. Generally, the NAQPMS adequately captured the observed spatiotemporal features of $O_3$ during the period of July 1st to August 31st in 2017 on the NCP. The results of the source apportionment indicated that the contributions of local emissions and transport from the NCP accounted for the largest proportion of $O_3$, with magnitudes of 25% and 39%, respectively. Compared with those in the average monthly results, the local contribution and regional transport during $O_3$ episodes on the NCP increased by 7% and 10%, respectively. Based on sensitivity tests, two thresholds of the sensitivity indicator $P(H_2O_2)/P(HNO_3)$ were detected, at 0.08 and 0.2. Ozone formation in the urban sites of Beijing, Tianjin, and the southern part of Hebei Province was controlled by VOCs, while the other sites were mainly controlled by $NO_X$. Biogenic emissions contributed approximately 18% to $O_3$ formation in July in the southwestern part of Hebei Province.

**Keywords:** $O_3$; NCP; source apportionment; sensitivity analysis; biological source contribution

## 1. Introduction

Ground-level ozone is a secondary pollutant that predominantly results from chain photochemical reactions involving nitrogen oxides ($NO_X$ = NO + $NO_2$), carbon monoxide (CO), and volatile organic compounds (VOCs) with the catalysis of sunlight in the troposphere [1,2]. Photochemical reactions contributed ~90% to global surface ozone, 9 times the contribution of the exchange from the stratosphere to the troposphere [3,4]. Involving findings have been comprehensively reviewed that high levels of ozone have adverse effects on human health [5], ecology [6], agricultural productivity [7,8], and potentially affect climate change [9].

In China, studies have typically prioritized acid rain, particulate matter, and more recently fine particulate matter ($PM_{2.5}$). With growing regional urbanization and industrialization, emissions involving chemical precursors have increased sharply, and field measurements have recently revealed very high concentrations of ozone in major Chinese city clusters. The projections revealed that ozone pollution conditions will worsen in the future [9]. The NCP, including Beijing, Tianjin, and Hebei, has a population of 200 million. As the center for economic and political development, the $O_3$ concentration in this area is reported to have increased from 2015 to 2018, with a growth rate of up to 22.84% [10]. Wang et al. pointed out that $O_3$ levels on the NCP exceeded the ambient air quality standard by 100–200% [11]. Therefore, information about ozone source apportionment and its controlling factors on the NCP is urgently needed for policy-makers in China. With the simulation of 3-D air quality models, Wang et al. [12] found that both local and regional sources played a significant role in ozone episodes in Beijing. Xiao et al. employed RAMS-CMAQ coupled with the ISAM module to research ozone episodes in June 2015 over the NCP and concluded that emissions sources in Shandong and Hebei were the major contributors to $O_3$ production [13]. Tagged simulations conducted by WRF-Chem suggested that local sources played an important role in the formation of the summer $O_3$ peak on the NCP, but that sources in Northwest China should not be neglected in controlling summer $O_3$ on the NCP [14]. Previous source apportionment studies mostly focused on specific episodes or the whole NCP, lacking more detailed studies considering the diurnal variations and the urban scale perspective in the source apportionment.

The complexity of ozone formation lies in the nonlinear dependence of $O_3$ production on its precursors, $NO_X$ and VOCs. Identifying the $O_3$ formation regime is the principal aspect of the science-based regulation of ozone pollution. Xu et al. [15] found in 2008 that urban sites in Beijing were VOC-limited and that rural sites were $NO_X$-limited, while Duan et al. [16] found that the ozone peak concentration in Beijing was limited by both $NO_x$ and VOCs. Observational analyses calculating $O_3$ sensitivity have also been conducted, such as Tang et al. [17], who used global ozone monitoring experiment (GOME) measurements over the NCP region (finding VOC-limited conditions in summer). Given the dramatic changes in emissions in North China, more studies determining the NOx-VOC limitations of the ozone formation regimes should be performed.

Apart from the $O_3$ formation regime, the impact of biogenic emissions is another important aspect of ozone pollution control on the NCP. Based on the MEGAN 2.1 model for BVOC emissions, Wang et al. [18] found that BVOC emissions contributed 75–77% of the total emissions in Beijing in summer. Mo et al. [19] further calculated that among all the BVOC species, isoprene, which is sensitive to high temperatures, contributed the most to ozone formation, accounting for 27% of the total VOC ozone generation potential. However, the current research on the contribution of BVOCs to ozone is mainly concentrated in southern China such as the Yangtze River Delta, and research on the NCP region is still relatively lacking.

In this paper, we focus on the NCP region, and the Nested Air Quality Prediction Modeling System (NAQPMS) coupled with an online source-tagged model was applied to quantify the $O_3$ source-receptor relationship, identify the ozone sensitivity threshold with $P(H_2O_2)/P(HNO_3)$ as the sensitivity indicator, and investigate the contribution of biogenic emissions to ozone formation. These results are beneficial for developing effective emergency and long-term control strategies for reducing the ambient $O_3$ level and improving air quality on the NCP.

## 2. Experiments

### 2.1. Model Description and Setup

A fully modularized three-dimensional Eulerian chemical transport model, the Nested Air Quality Prediction Model System (NAQPMS) developed by the Institute of Atmospheric Physics, Chinese Academy of Sciences [20], was utilized in this study. NAQPMS can reproduce the physical and chemical evolution of reactive pollutants in the ambient atmosphere by solving the mass balance equation for

terrain-following coordinates. The terrain-following coordinate [21] has advantages in implementing boundary conditions and consider the terrain influence. The carbon-bond mechanism Z (CBM-Z) [22], composed of 71 species and 134 chemical reactions, was implemented in NAQPMS to simulate gaseous chemistry. The RADM mechanism involving 22 species was employed to describe the aqueous chemistry and wet deposition process. The inorganic aerosol chemistry and secondary organic aerosol chemistry in NAQPMS were addressed with the ISSOROPIA1.7 model [23] and a bulk two-product yield parameterization [24], respectively. An accurate radiative-transfer model (TUV version 4.5) with an eight-stream discrete ordinate solver was coupled online with NAQPMS to evaluate the effects of aerosols on photolysis frequencies and tropospheric oxidants [25]. For heterogeneous chemical processes, 12 species and 28 reactions involving dust, sea salt, sulfate, and black carbon were included, and the updated model has recently been successfully applied to simulate mixing between dust and anthropogenic gaseous pollutants in East Asia [26]. The advection scheme used an accurate mass-conservative, peak-preserving algorithm [27]. The dry deposition velocity was parameterized using a scheme proposed by Wesely [28]. The meteorological input for the NAQPMS was provided by the Weather Research and Forecasting Model (WRF) [29], a new generation of prediction models and assimilation systems developed by a joint effort of the National Center for Atmospheric Research (NCAR), the National Centers for Environmental Prediction (NCEP) and other institutes and colleges. WRF has been widely applied in the fields of meteorological research and numerical weather prediction, and it can be used not only to simulate real cases but also to research fundamental physical processes with a single module.

As shown in Figure 1a, the modeling domain centered on Beijing in this study included three nested domains covering all of China, the North Plain, and the NCP region. The horizontal grid resolutions are 27 km for domain 1 (D1) with 148 × 121 grids, 9 km for domain 2 (D2) with 136 × 124 grids, and 3 km for domain 3 (D3) with 178 × 244 grids. Vertically, the model employed terrain-following coordinate, and the vertical grid spacing was divided into 30 layers increasing gradually from 20 m near the surface to 20 km at the top with 20 layers in the boundary layer. The simulation period was from July 2017 to August 2017, and the first 3 days were used as the spin-up period to reduce the influence of the initial conditions.

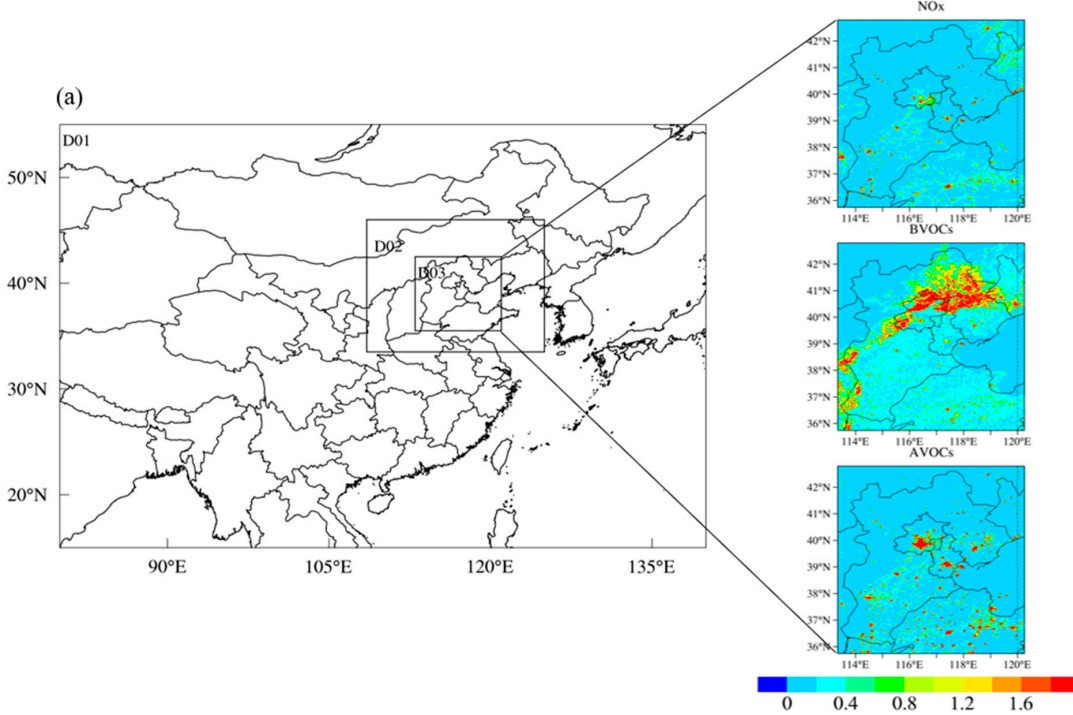

**Figure 1.** *Cont.*

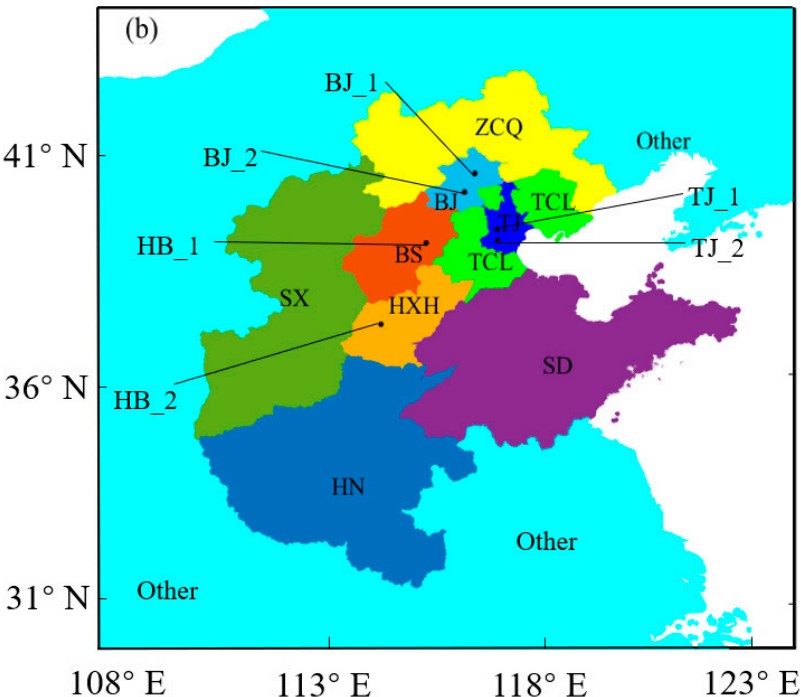

**Figure 1.** (**a**) Model domain for NAQPMS with monthly average $NO_X$, BVOCs, AVOC emission rates (in $\mu g/m^3/s$) and (**b**) The interior of domain3 with trace tagged regions. The solid circles represent the 6 observation sites: BJ_1 (40.14° N, 116.72° E); BJ_2 (39.98° N, 116.40° E); TJ_1 (39.17° N, 117.145° E); TJ_2 (39.03° N, 117.71° E); HB_1 (38.87° N, 115.52° E); HB_2 (37.10° N, 114.48° E).

*2.2. Emission Inventory*

The emissions data used in this study included an anthropogenic emissions inventory and biomass inventory data. The anthropogenic emission inventory was obtained from MEIC (Multi-resolution emission inventory for China) developed by Tsinghua University (http://www.meicmodel.org/index. html, accessed on 19 November 2019) based on 2016 with a resolution of 0.1° × 0.1°. The MEIC was used in this study to investigate the emission responses to ozone concentration. MEIC is a dynamic technology-based inventory developed for China covering the years from 1990 to recent years by Tsinghua University [30] covering 10 major air pollutants and greenhouse gases ($SO_2$, NOx, CO, NMVOC, $NH_3$, $CO_2$, $PM_{2.5}$, $PM_{10}$, BC, and OC) and more than 700 anthropogenic emission sources. Considering that China has been vigorously controlling pollution in recent years, the MEIC inventory includes recent control policies based on the available official reports to provide a reliable description of pollutant emissions. In general, the MEIC is a technologically advanced, reliable, and widely used inventory. Exactly, the latest MEIC has covered the year of 2016, but the data updated in the website is lagging, we, fortunately, get the data collection by personal contact and applied it to this study. Five source categories, i.e., residence, agriculture, industry, power plant and transport, and 25 species of pollutants were included. The largest NOx hotspot is in the main urban region of Beijing, where vehicle exhaust emissions are high. Biogenic emissions were obtained from a biogenic emission model named MEGANv2 (Model of Emission of Gases and Aerosols from Nature) developed by NCAR (http://accent.aero.jussieu.fr/database_table_inventories.php, accessed on 21 December 2019). This model is driven by meteorological fields, plant functional types, leaf area index, and emission factors classified by VOC species. More details can be found in the study of Wang et al. [18]. For the areas of high vegetation coverage in the Yanshan Mountains and Taihang Mountain over the NCP, as shown in Figure 1a, the concentration of BVOC (Biogenic volatile organic compounds) was high in the northern part of the research area; this distribution pattern was the opposite of that for NOx. Anthropogenic volatile organic compounds known as AVOCs (the involving species considered in

this study included OLEI (Internal olefin carbons (C = C)), OLET (Terminal olefin carbons (C-C)), PAR (Paraffin carbon), TOL (Toluene) and XYL (Xylene)) are mainly produced by four categories of activity: transport, solvent use, production and storage processes, and combustion processes [31]. Their distribution is similar to that of NOx emissions. In addition, the magnitude of AVOCs is low compared to that of BVOCs, suggesting the non-negligible effect of BVOCs on ozone formation.

## 2.3. Source Apportionment

To qualitatively and quantitatively evaluate the ozone source-receptor relationship on the NCP, an online source apportionment module that attributes pollutants to different geographical locations at each step of the simulation without influencing the standard calculations was implemented in NAQPMS. Compared with the classic sensitivity analysis that turns on and off emissions in targeted ozone production regions, the tagged tracer method provides a different and more efficient measurement of the relative importance of various ozone production regions and lacks the errors introduced by important non-linearities in the transport and fast photochemistry of ozone and its precursors. A detailed description of the methods of the trace tagging module can be found in previous studies [32,33]. In this study, the ten ozone production regions were tagged mainly according to administrative divisions, as were the initial conditions and lateral conditions. As shown in Figure 1b, the 10 source regions are Beijing (BJ); Tianjin (TJ); the Hebei region, divided into Shijiazhuang-Baoding (BS), Zhangjiakou—Chengde–Qinhuangdao (ZCQ), Tangshan–Cangzhou–Langfang (TCL), and Hengshui–Xingtai–Handan (HXH); three source regions outside the NCP (NCP), Shandong (SD), Shanxi (SX), Henan (HN), and other regions (OTH).

## 2.4. Model Evaluation

The NAQPMS is driven by the mesoscale numerical weather prediction system (WRF), and the simulation of the meteorological field is vital to the simulation of pollutants, especially to the species sensitive to temperature. Therefore, we evaluate the performance of WRF at 6 stations consisting of 3 rural sites and 3 urban sites with the MICAPS data by calculating several important statistical parameters in Table 1. We can see that WRF rationally recaptured the magnitudes of the air temperature with R up to 0.9, and the MB and the RMSE of most sites are around 4 °C and 5 °C, providing sufficient confidence for the following discussion.

**Table 1.** Statistical parameters of the Weather Research and Forecasting Model (WRF) simulated air temperature in six stations in July and August 2017.

| Station | Longitude (°E) | Latitude (°N) | R | MO | MP (°C) | MB (°C) | RMSE | NMB (%) |
|---|---|---|---|---|---|---|---|---|
| Beijing | 116.28 | 39.93 | 0.9 | 26.6 | 31.4 | 4.9 | 5.5 | 18.3 |
| Chaoyang | 116.48 | 39.95 | 0.9 | 27.1 | 31.2 | 4.1 | 4.2 | 15.3 |
| Haidian | 116.28 | 39.98 | 0.9 | 26.0 | 31.5 | 5.5 | 4.7 | 21.0 |
| Huairou | 116.63 | 40.32 | 0.9 | 25.5 | 30.3 | 4.7 | 5.7 | 18.5 |
| Changping | 116.22 | 40.22 | 0.9 | 26.3 | 31.0 | 4.6 | 5.4 | 17.6 |
| Shijingshan | 116.18 | 39.93 | 0.9 | 26.1 | 31.3 | 5.8 | 4.7 | 20.0 |

To evaluate the performance of NAQPMS in both urban and rural regions of the NCP, both urban and rural sites were chosen in Beijing city, Tianjin city, and Hebei Province. Figures 2 and 3 presented the observed and simulated hourly surface ozone and $O_x$($O_3$ + $NO_2$) at the six selected sites over the NCP. The observed data from 1 June 2017 to 31 August 2017 were provided by the China National Environment Monitoring Centre (CNEMC). In general, the NAQPMS rationally recaptured the magnitudes and trends of the ozone pattern and was comparable to previous modeling studies in this region [25]. The correlation coefficients (R) between the observed and simulated ozone and $O_X$ were 0.56 and 0.58, respectively. A total of 66.36% of the simulated $O_3$ values were within a factor of

0.5 to 2 of the observations (FAC2), while 88.15% of the simulated $O_x$ values were within the FAC2 (Figure 4). The statistical parameters of NMB, NME, and RMSE for $O_3$ ranged from 0.22–0.62, 0.4–0.71, 55.74–84.1, respectively; and the above indicators for $O_X$ ranged from 0.11–0.32, 0.27–0.4, and 48.55 to 65.13, respectively. Quantitatively, the simulated ozone concentration overestimated the observations on 22–24 July and 27–28 August. This overestimation is widespread in the current atmospheric transport model in East Asia [34,35]. Studies in MCIS-Asia III revealed that this underestimation is mostly attributed to uncertainties in the treatment of the heterogeneous chemistry and vertical transport in boundary layers in the current CTMs [35].

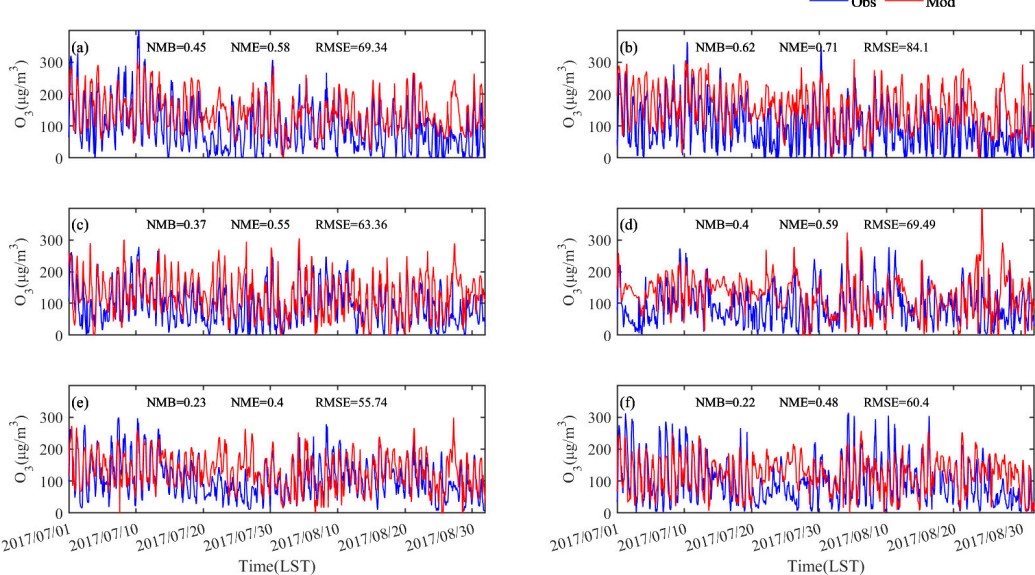

**Figure 2.** Observed (blue line) and simulated (red line) hourly mean surface $O_3$ in July and August 2017; (**a**–**f**) represent the observation sites BJ_1, BJ_2, TJ_1, TJ_2, HB_1, and HB_2, respectively.

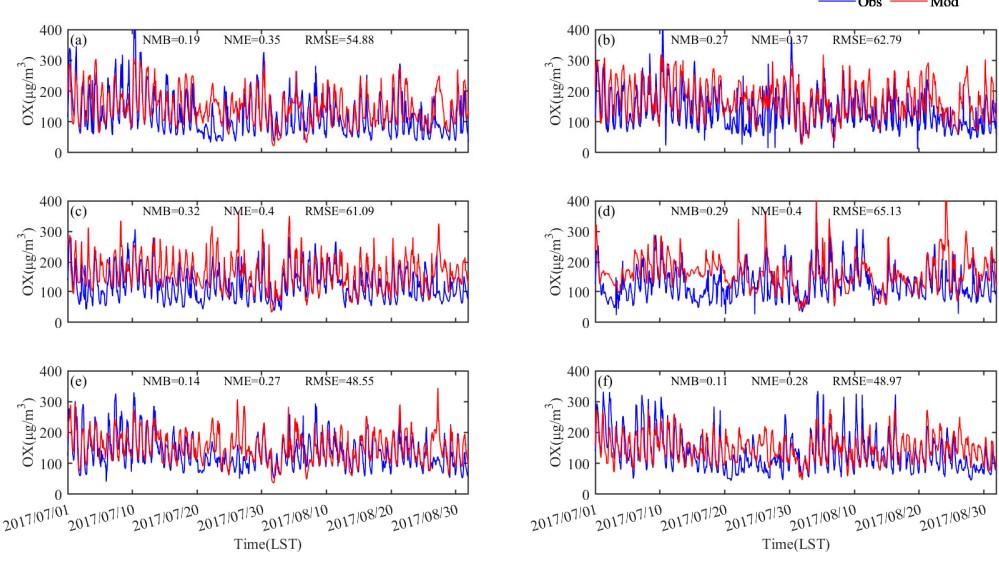

**Figure 3.** Observed (blue line) and simulated (red line) hourly mean $O_X(O_3 + NO_2)$ in July and August 2017; (**a**–**f**) represent the observation sites BJ_1, BJ_2, TJ_1, TJ_2, HB_1, and HB_2, respectively.

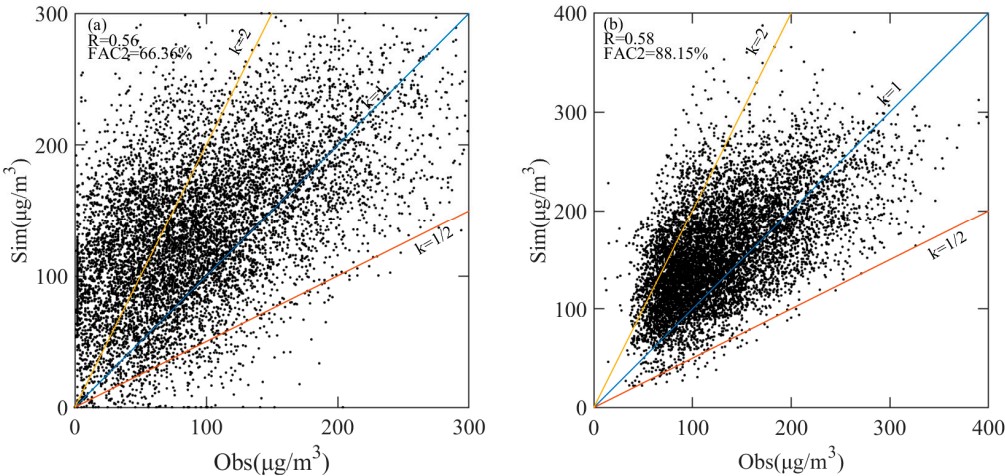

**Figure 4.** Scatter plots for the comparison of observed and simulated $O_3$ (**a**) and $O_X$ (**b**) with the statistical parameters R and FAC2; the 2:1 line (yellow), 1:1 line (blue), and 1:1/2 line (red) are also shown.

As the ozone formation is temperature-dependent, we also simply analyzed the impact of temperature uncertainty inert of the WRF model on the ozone simulation. As shown in Figure 5, the averaged observed ozone generally increased with the increasing air temperature, and the averaged simulated and observed air temperature are 31 °C and 26 °C, respectively. The discrepancy caused by uncertainty in the air temperature of ozone is about 30 μg/m$^3$, which is within an acceptable range.

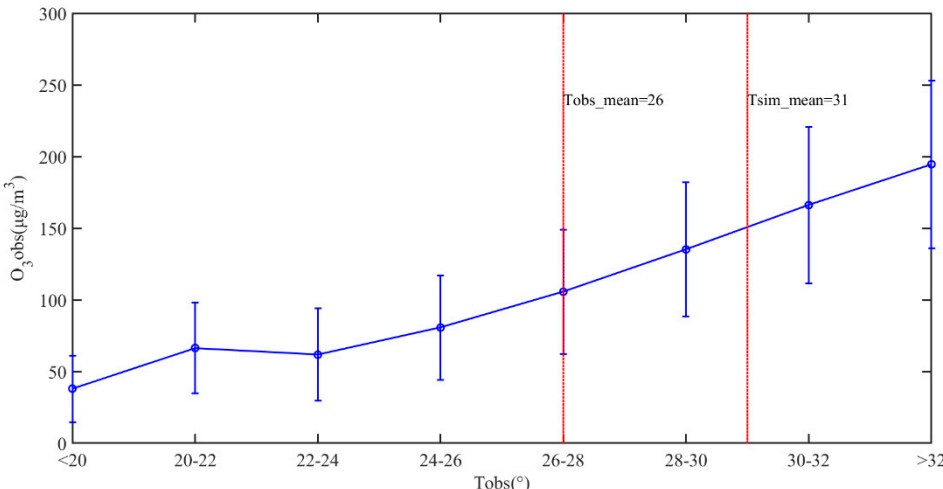

**Figure 5.** The observed $O_3$ concentration under different levels of observed air temperature with the range of 20–32 °C; the two red lines represent the averaged observed air temperature and averaged simulated temperature, respectively.

## 2.5. Indicator of Ozone-Sensitivity Regime

In this study, the $P(H_2O_2)/P(HNO_3)$ ratio was selected as the sensitivity indicator for studying the ozone formation regimes over the NCP. Ozone formation results from a chain of nonlinear chemistry reactions that are terminated by the cross-reactions of $RO_X$ and/or $NO_X$ depending on the abundance of $NO_X$. Under high $NO_X$ conditions, the termination process is dominated by the reactions of $NO_2$ with OH and $RO_2$, as shown in reaction (R1)–(R2), and this process finally generates so-called $NO_Z$ species, the rate constant for reactions R1, R2, R3 can also be found in the reference [22]. NAQPMS is capable of calculating and outputting the values of OH and $HO_2$ online for each time step and each grid without assumptions. Under low $NO_X$ conditions, the dominant processes are self-reactions of $HO_2$(R3) and cross-reactions of $HO_2$ and $RO_2$(R4), resulting in the formation of hydrogen peroxides ($H_2O_2$) and organic peroxides. Hence, the ratio of $P(H_2O_2)/P(HNO_3)$ can reflect the ambient atmospheric conditions; a higher $P(H_2O_2)/P(HNO_3)$ ratio indicates a $NO_X$-limited regime, while a lower $P(H_2O_2)/P(HNO_3)$ ratio corresponds to a VOC-limited regime. This indicator has been applied in other studies [16,36].

$$OH + NO_2 \rightarrow HNO_3 \tag{R1}$$

$$RO_2 + NO_2 \rightarrow RO_2NO_2 \tag{R2}$$

$$HO_2 + HO_2 \rightarrow H_2O_2 + O_2 \tag{R3}$$

$$HO_2 + RO_2 \rightarrow RO_2H + O_2 \tag{R4}$$

## 3. Results

### 3.1. Spatial Distribution of the Simulated Ozone

Figure 6 illustrates the simulated monthly average concentrations of $O_3$ over the NCP during the day (LST: 6:00–18:00) and night (LST: 18:00–6:00) in July and August. During the day, the highest concentrations, 190 $\mu g/m^3$, were simulated with NAQPMS in the majority of Beijing in July, and the high ozone levels (>170 $\mu g/m^3$) extended hundreds of kilometers north in Hebei Province. The simulated $O_3$ concentration in August was generally lower than that in July, with a maximum value of 170 $\mu g/m^3$. At night, high concentrations of $O_3$ occurred over the western part of Beijing and in the northern and western parts of Hebei Province in July. In August, the simulated $O_3$ distribution pattern was similar to that in July; however, the magnitudes of $O_3$ were lower. In both August and July, the daytime ozone concentration was simulated to be higher than that at night, which can be explained by the chemical reactions involving nitrogen oxide (NO) that occur at night and the physical process of dry deposition that results in the depletion of ozone [37]. There was little discrepancy in the simulated $O_3$ between day and night in the northern part of the NCP, where the $O_3$ concentration is high in July, which indicates a weak NO titration reaction in this region. In contrast, more serious pollution with a high concentration of NO occurred in the southern part of the NCP, resulting in a large gap between daytime and nighttime $O_3$ levels in the simulation.

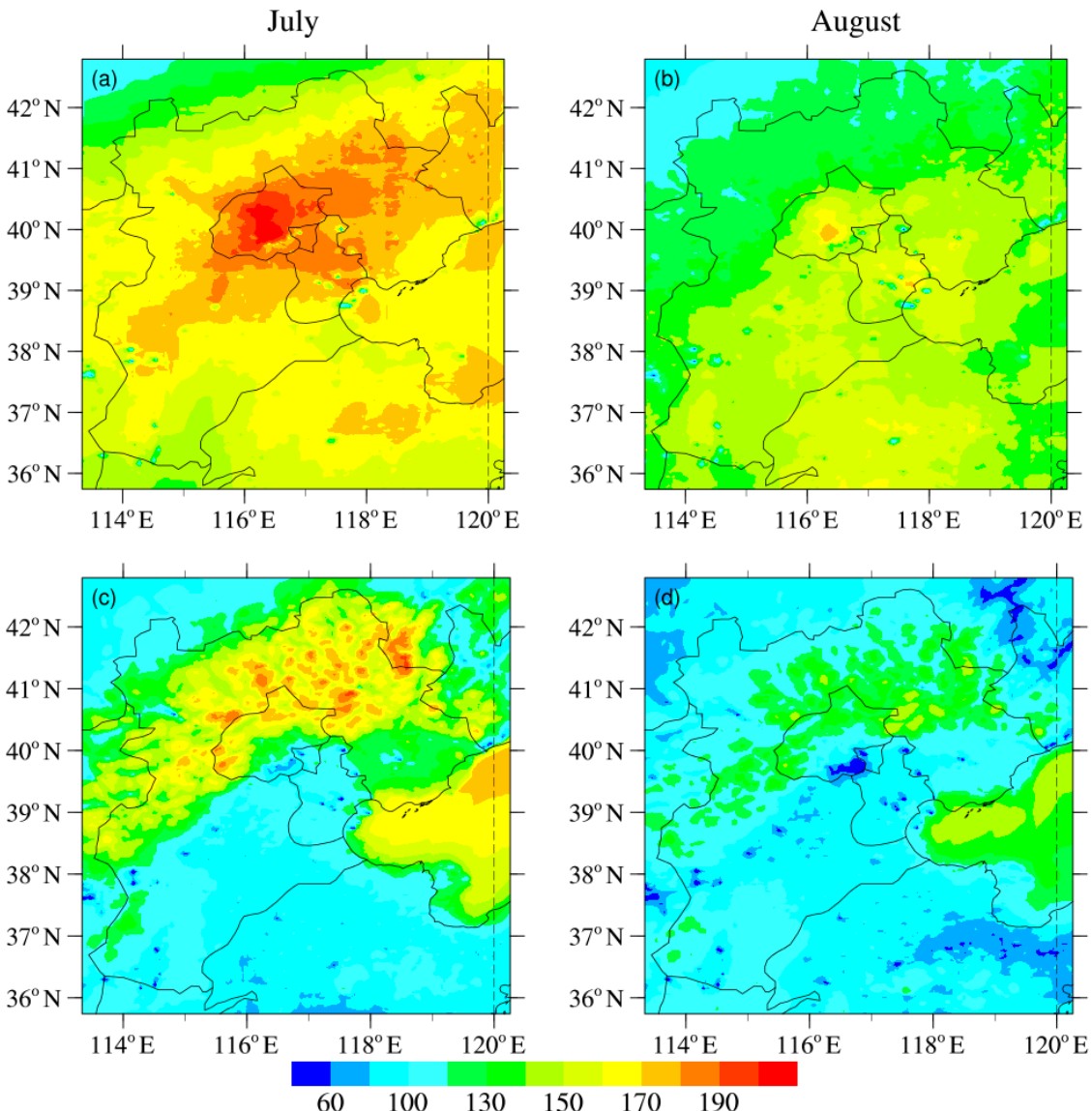

**Figure 6.** Simulated monthly average surface O₃ distribution (in µg/m³) over the NCP (**a–d**); (**a,c**) display July, (**b,d**) display August, and the rows from top to bottom display conditions during the day (LST: 6:00–18:00) and night (LST: 18:00–6:00), respectively.

*3.2. Analysis of Ozone Source Apportionment*

3.2.1. O₃ Source Apportionments for 13 Cities on the NCP

To highlight the sources for these O₃ episodes, source apportionments for the monthly mean ozone and ozone-exceedance days (O₃ > 160 µg/m³) in 13 cities on the NCP are shown in Figure 7. In general, the contributions of local emissions and transport from the NCP accounted for the largest proportion of O₃ during the episodes in most cities except for Tangshan and Qinhuangdao. When O₃ episodes occurred, the average local contributions and the contributions from the NCP for the 13 cities are 32% and 49%, respectively. Compared to the results of monthly averaged source apportionments shown in Figure 7a, during O₃ episodes, the contributions of local emissions and transport from the NCP increased by up to 7% and 10%, respectively. For further analysis, during O₃ episodes, the self-contribution, the contribution from neighboring cities within the NCP region and the contribution from Henan, Shandong, and Shanxi to Beijing were 43%, 22%, and 10%, respectively, while on clean days, the values were 31%, 22%, and 10%, respectively, showing that the effect on Beijing decreases with increasing

distance. Beijing's terrain is surrounded by mountains on three sides, which favors the accumulation of local $O_3$. Additionally, the emissions of precursors involving $NO_X$ and VOCs are high, as shown in Figure 1. The transport from neighboring Henan Province also plays an important role in the $O_3$ levels in Xingtai and Handan regardless of whether an $O_3$ episode is occurring, revealing the effect of prevailing southerly winds in these regions.

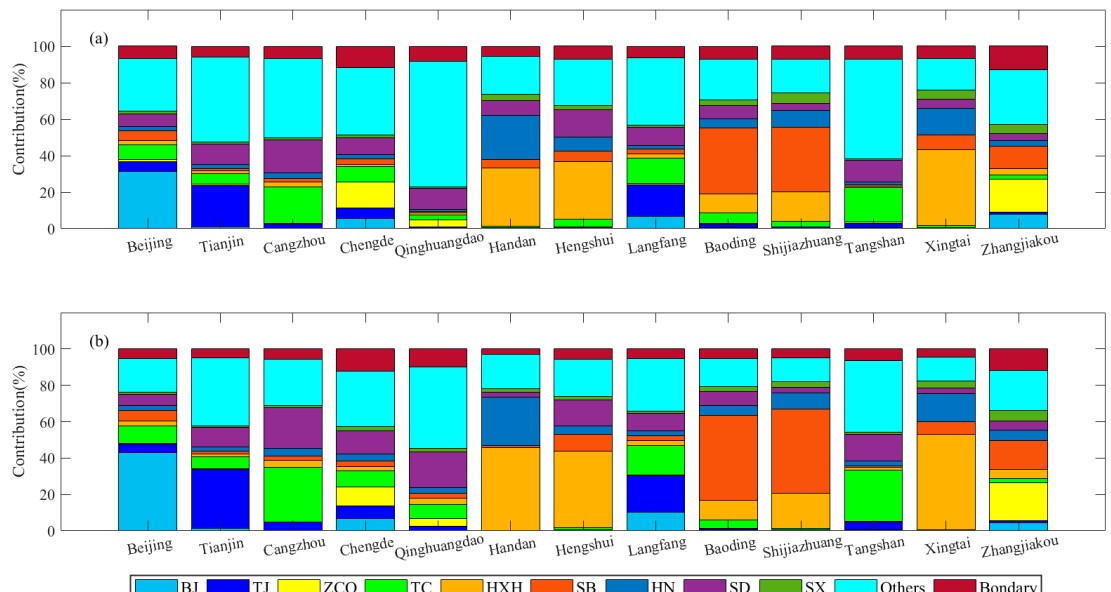

**Figure 7.** $O_3$ spatial source apportionments by source regions in cities on the NCP (**a**) monthly average results; (**b**) during $O_3$ episodes.

In addition to analyzing the spatial contributions, we also quantified the temporal source contributions. Figure 8 presents the temporal source apportionment contributions for the 13 cities within the NCP region during episodes and on clean days. Day 0 represents contributions from the "current day", while day 1 and day 2 denote contributions from one day and two days prior to day 0, respectively. Among the 13 cities, the contribution of the "current day" of most cities exceeded 50%, and the maximum value found in Beijing exceeded 60%. This indicates that contributions on the day of emission contribute the most to $O_3$ formation, with average values of 44% and 46% on clean and episode days, respectively, and the contribution of the "current day" on episode days is slightly higher than that on clean days. The contributions on the day before the day of emission are the second-largest source, with average values of 27% and 28% on clean and episode days, respectively, revealing that the temporal contribution decreases over time. In summary, the local source emissions and transport from neighboring areas of the "current day" are the dominant factors affecting $O_3$ formation on the NCP. These results indicate that policymakers should respond rapidly to $O_3$ episodes, and both regional collaboration and stricter local measures for emissions control should be considered to meet air quality standards and reduce episodes of high $O_3$ levels.

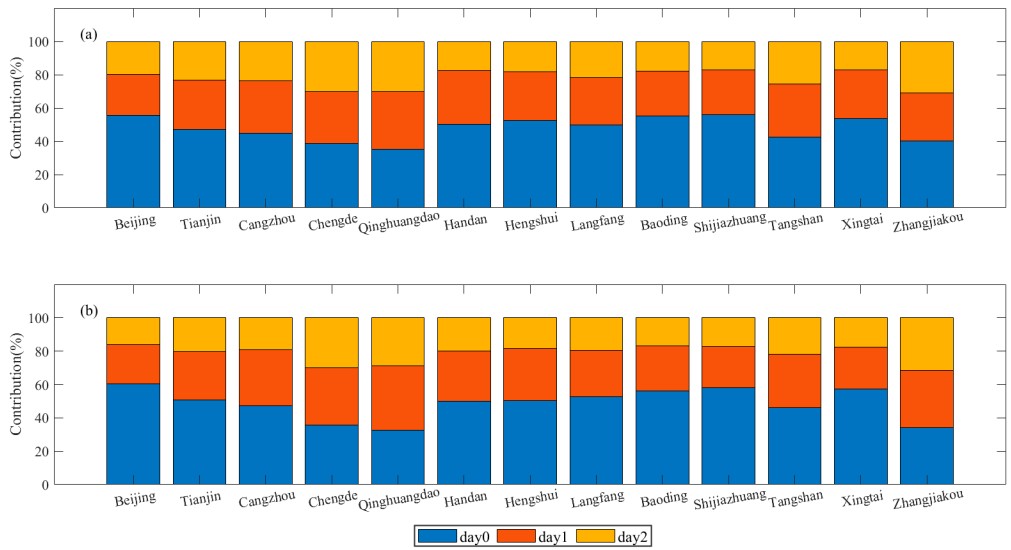

**Figure 8.** O$_3$ temporal source apportionments by source regions in cities on the NCP (**a**) monthly average results; (**b**) during O$_3$ episodes (day 0: contributions on the day of emission; day 1: emissions on the previous day; day 2: emissions two or more days earlier).

### 3.2.2. Diurnal Variation in O$_3$ Source Apportionments at 6 Sites in the NCP Region

Figure 9 shows the diurnal variation in the simulated concentrations of O$_3$ at 6 selected sites that were evaluated and their source apportionment averaged for July and August in 2017. In general, the diurnal variation in the O$_3$ concentration presented a single peak, with a maximum observed in the afternoon. This peak was caused by the complex photochemical reactions occurring in the presence of sufficient solar radiation. The minimum was observed in the morning when the O$_3$ concentration fell below 30 μg/m$^3$. Among the 6 selected sites, the O$_3$ concentration at the two sites in Beijing was the highest, and the O$_3$ concentration simulated in urban sites was higher than that in the rural sites in Beijing and Tianjin.

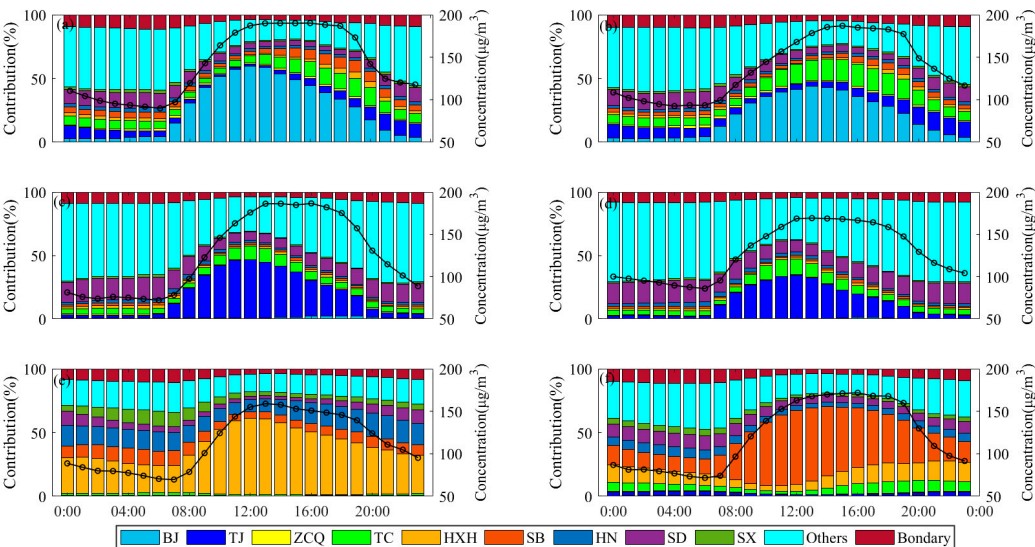

**Figure 9.** Diurnal variation in O$_3$ concentration (black lines with circles, right axis) and its source apportionment by region (bars, left axis) at 6 sites; (**a–f**) represent the observation sites BJ_1, BJ_2, TJ_1, TJ_2, HB_1, and HB_2, respectively.

Generally, self-contributions accounted for approximately 40% and 60% of the ozone at rural and urban sites in Beijing in the afternoon. Variations in self-contribution were similar to those of the ozone concentration, revealing the significance of local emissions during the day. At night, when the $O_3$ concentration was low, transport from other regions both within the NCP and outside of the NCP played an important role. In Tianjin, regional transport from emissions outside the NCP occupied a larger proportion than in the sites in Beijing and Hebei Province, accounting for approximately 60% in the early morning, while a self-contribution below 35% was simulated in the rural sites even if the $O_3$ concentration reached a peak. Influenced by the prevailing southerly winds in the summer, the contribution of transport from regions upwind of Shijiazhuang, Baoding, Tangshan, Cangzhou, and Langfang played an important role in the source apportionment of the two sites in Hebei Province. In summary, at the two sites in Hebei Province, the variation in self-contribution maintained an analogous pace with that of $O_3$ concentration, with the maximum value exceeding 50%.

### 3.3. Ozone Sensitivity Analysis

3.3.1. Ozone Sensitivity Analysis over the NCP

To understand the ozone precursor relationships over the NCP, we performed the following sensitivity tests to determine the thresholds of the selected ozone formation indicator $P(H_2O_2)/P(HNO_3)$. $P(H_2O_2)/P(HNO_3)$ was chosen because this ratio in particular is closely related to the chemical factors that create the division between NOx-sensitive and VOC-sensitive conditions [38].

The sensitivity experiment design scheme is shown in Table 2. In the Base condition, the baseline emission source was used, while in Case 1 and Case 2, a 50% reduction in the $NO_X$ emission source and a 50% reduction in the VOC emission source were used, respectively, in the NAQPMS to calculate the ozone formation rate ($P(O_3)$), which is defined as follows:

$$P(O_3) = \frac{C_N - C_B}{C_B} \times 100\% \tag{1}$$

$$P(O_3) = \frac{C_V - C_B}{C_B} \times 100\% \tag{2}$$

where $C_N$ and $C_V$ denote the simulated ozone concentrations in Case 1 and Case 2, respectively, and $C_B$ denotes the simulated ozone concentration when using the baseline emission source in the model. Figure 10 is the scatter plot of the ozone formation indicator ($P(H_2O_2)/P(HNO_3)$) and the ozone formation rate ($P(O_3)$). It should be mentioned that the extracted data were all urban grid points, and the extraction time is the period when the ozone concentration was high (11–13 July 2017 and 8–10 August 2017) to better reflect the relationships between the ozone formation rate and the ozone formation sensitivity indicator. We can see that in Case 1, when NOx emissions were reduced by 50%, the ozone formation rate increased with the increasing $P(H_2O_2)/P(HNO_3)$ ratio, revealing that the ozone formation regime transformed from a $NO_X$-limited regime to a VOC-limited regime; in contrast, the ozone formation regime transformed from a VOC-limited regime to a $NO_X$-limited regime when the VOC emissions were reduced by 50%. The intersections of the ozone formation rate and the ozone formation indicator shown in Figure 10 are the thresholds for inferring the ozone formation regime, and two values, 0.08 and 0.2, were detected by correlation calculation. We can conclude that ozone was controlled by VOCs when the ratio of $P(H_2O_2)/P(HNO_3)$ was below 0.08, ozone was controlled by VOCs when the ratio of $P(H_2O_2)/P(HNO_3)$ was above 0.2, and ozone was controlled by both $NO_X$ and VOCs when the ratio was in the range of 0.08 to 0.2, and the results obtained from this study agree well with the results from the research of Tonnesen and Dennis [39], the thresholds of which are 0.06 and 0.2. Except for the ratio of $P(H_2O_2)/P(HNO_3)$, several other indicators have been developed to diagnose the $O_3$-precursor relationship, we have reviewed other photochemical indicators (NOy, HCHO/NO$_2$) and compared them to our results. The transition point between NOx- and VOC-sensitive locations for the NCP occurred at approximately 7.87 ppb NOy, which is close to that for the Barcelona area [40]

and San Joaquin Valley [41]. However, the NOy transition value in our study is higher than the result conducted in Lake Michigan, which is considered a scenario with little or no biogenic VOCs [42]. For the ratio of $HCHO/NO_2$, Tonnesen and Dennis [40] found that in situ measurements of the ratios of which between 0.8 and 1.8 indicating a "transition" environment where $O_3$ was both sensitive to radicals and $NO_X$, and the two thresholds we calculated are 0.9 and 2.2. Considering the complexity of the real ambient air, we are supposed to note that differences and uncertainties still exist in applying the indicators to determine the ozone formation regime and thresholds often vary by time and region.

**Table 2.** Sensitivity experiment design scheme—Base: baseline emission source; Case 1: 50% reduction in the NOx emission source; Case 2: 50% reduction in the VOC emission source.

|        | $NO_X$ | VOCs |
|--------|--------|------|
| Base   | 100%   | 100% |
| Case 1 | 50%    | 100% |
| Case 2 | 100%   | 50%  |

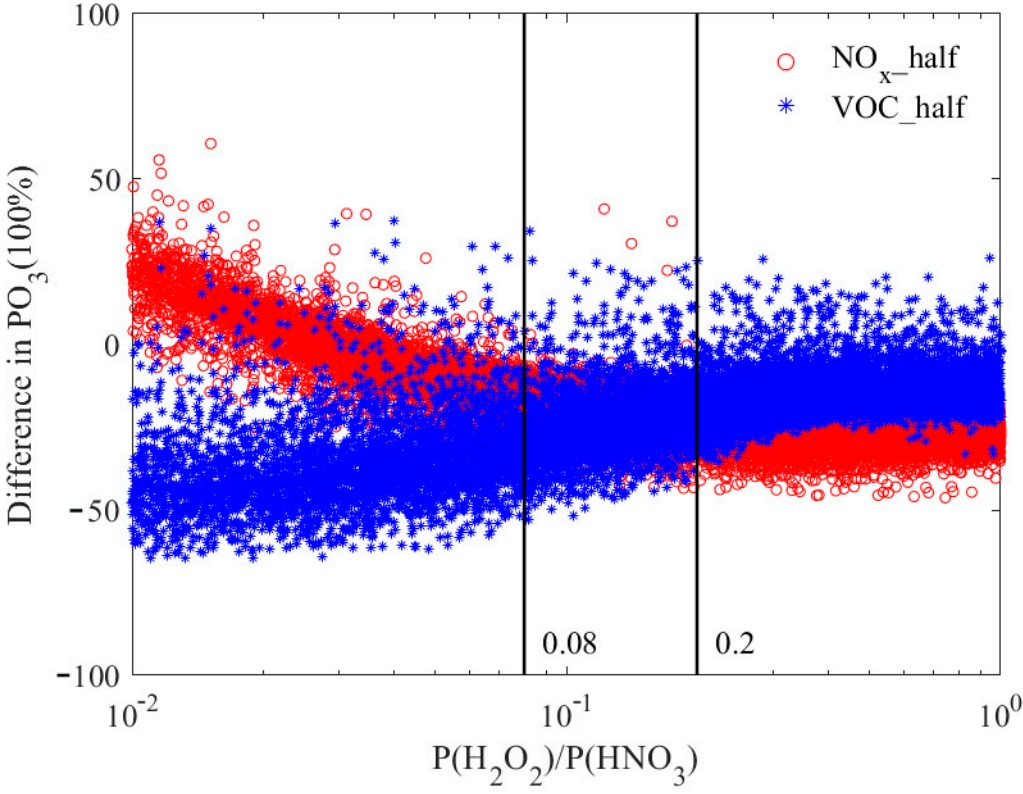

**Figure 10.** Scatter plot for the ratio of $P(H_2O_2)/P(HNO_3)$ and the difference in $P(O_3)$; blue indicates the sensitivity test with a 50% reduction in NOX emissions, and red indicates the sensitivity test with a 50% reduction in VOC emissions.

Figure 11 illustrates the monthly distribution of ozone-precursor relationships over the NCP with the indicator discussed above at different ozone levels. On clean days, when the ozone concentration was below 100 μg/m³, the majority of Beijing city and Tianjin city and several cities lying in the southwestern part of Hebei province experienced a VOC-limited regime, while the northern rural part of Beijing, Chengde city and Zhangjiakou city were controlled by NOx. The transition regime was small. In episodes when the ozone concentration was above 200 μg/m³, a transition in $O_3$ production from a VOC-limited to a $NO_X$-limited regime occurred near the urban area, which can be attributed to the poor meteorological diffusion conditions and high levels of pollutant emissions from urban areas. The precursors transported to the nearby regions corresponded to the change in the dominant factors

that affect ozone formation. Furthermore, we also found that the opposite conditions occurred in other locations, which may be explained by the control measures in the local region. Compared to that in July, the ozone formation in August was more controlled by VOCs. On clean days, the southern part of Beijing was almost entirely controlled by VOCs, and Baoding, Shijiazhuang, Handan, and Xingtai were transformed into a VOC-limited regime.

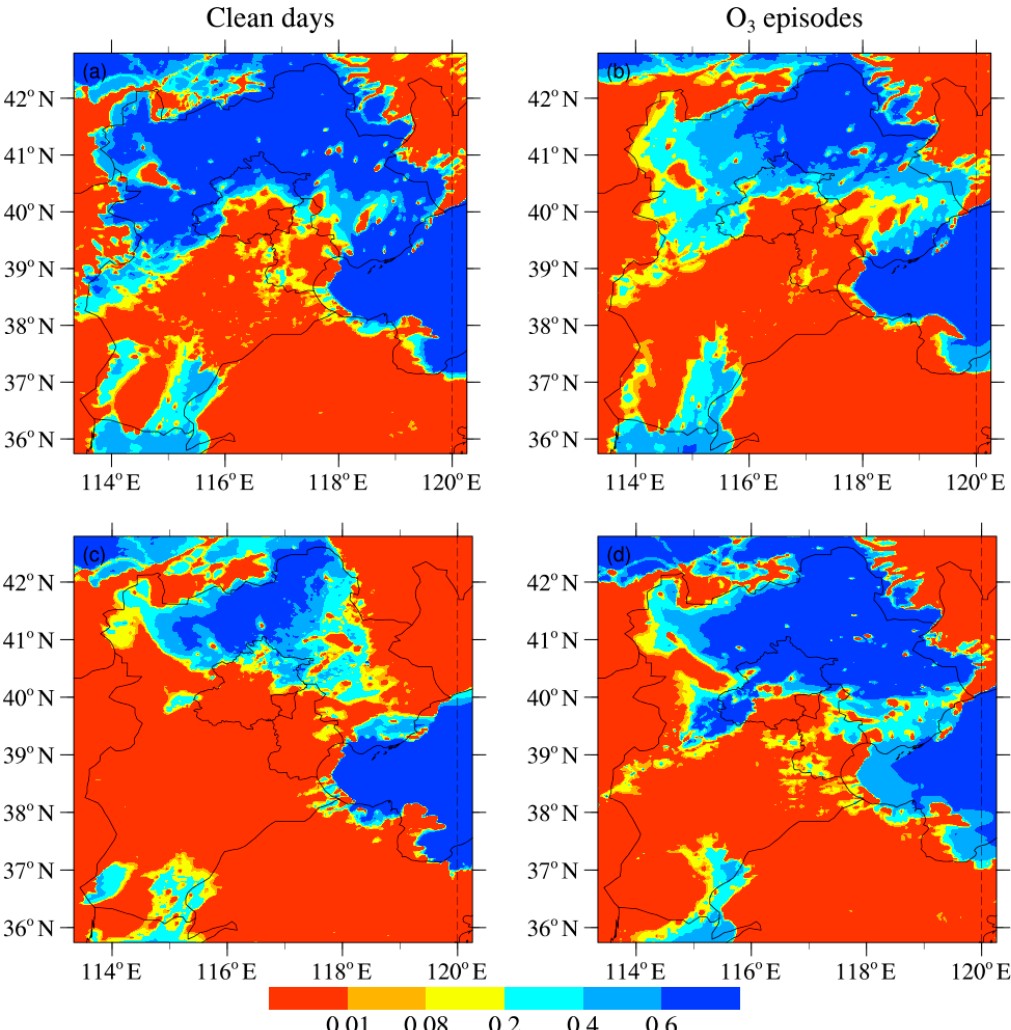

**Figure 11.** The distribution of the ratio of $P(H_2O_2)/P(HNO_3)$ in July and August 2017 over the NCP (**a**–**d**); (**a**,**c**) represent clean days ($O_3 < 100 \ \mu g/m^3$), (**b**,**d**) represent $O_3$ episodes ($O_3 > 200 \ \mu g/m^3$), and the rows from top to bottom represent July and August.

### 3.3.2. Ozone Net Production Analysis over the NCP

We also examined the net chemical production of $O_3$ (chemical production-chemical loss) within the boundary layer over the NCP. Here, the net chemical production, $N(O_3)$, is calculated by the equation, $N(O_3) = F(O_3) - D(O_3) = \{k_1[HO_2][NO] + k_2[RO_2][NO]\} - \{k_3[O(^1D)][H_2O] + k_4[OH][O_3] + k_4[HO_2][O_3] + k_5[O_3][olefin]\}$ in NAQM. The NAQPMS models give the net chemical production as the difference of $O_3$ mixing ratio between the calculation steps of the chemistry module with a process analysis package. The net chemical production was calculated in each grid and then the average was taken for all the selected grids. As shown in Figure 12, in general, the maximum net chemical production of ozone was simulated to be approximately 50 $\mu g/m^3$ over the urban sites of Beijing, the eastern part of Tianjin, and the southwestern part of Tangshan; meanwhile, negative ozone net chemical production was simulated over the sites adjacent to the urban and rural regions. Compared

to the maximum 8-h ozone net chemical formation, the high-value zone of the maximum 1-h zone net chemical production was distributed over a larger scale. This phenomenon is particularly obvious in August, which can be attributed to the rapid photochemical reactions during periods with strong solar radiation. The net chemical production of $O_3$ in this paper was averaged throughout the day, and the net chemical production during the day may be higher than the net chemical production at night. Some studies on $O_3$ net chemical production have been conducted in Europe and North America. For example, Connor et al. [43] estimated that the daytime net ozone production in the boundary layer over central Europe during July/August 2000 was within a range of 27.87–83.62 μg/m$^3$, and our results are consistent with this research. By the further analysis shown in Figure 1 (NOx, AVOC), we found that the pattern of ozone net production was similar to that of anthropogenic emissions, and pollution sources from anthropogenic emissions played a dominant role in ozone net chemical production, revealing the significant effect of anthropogenic source emissions on the model simulation.

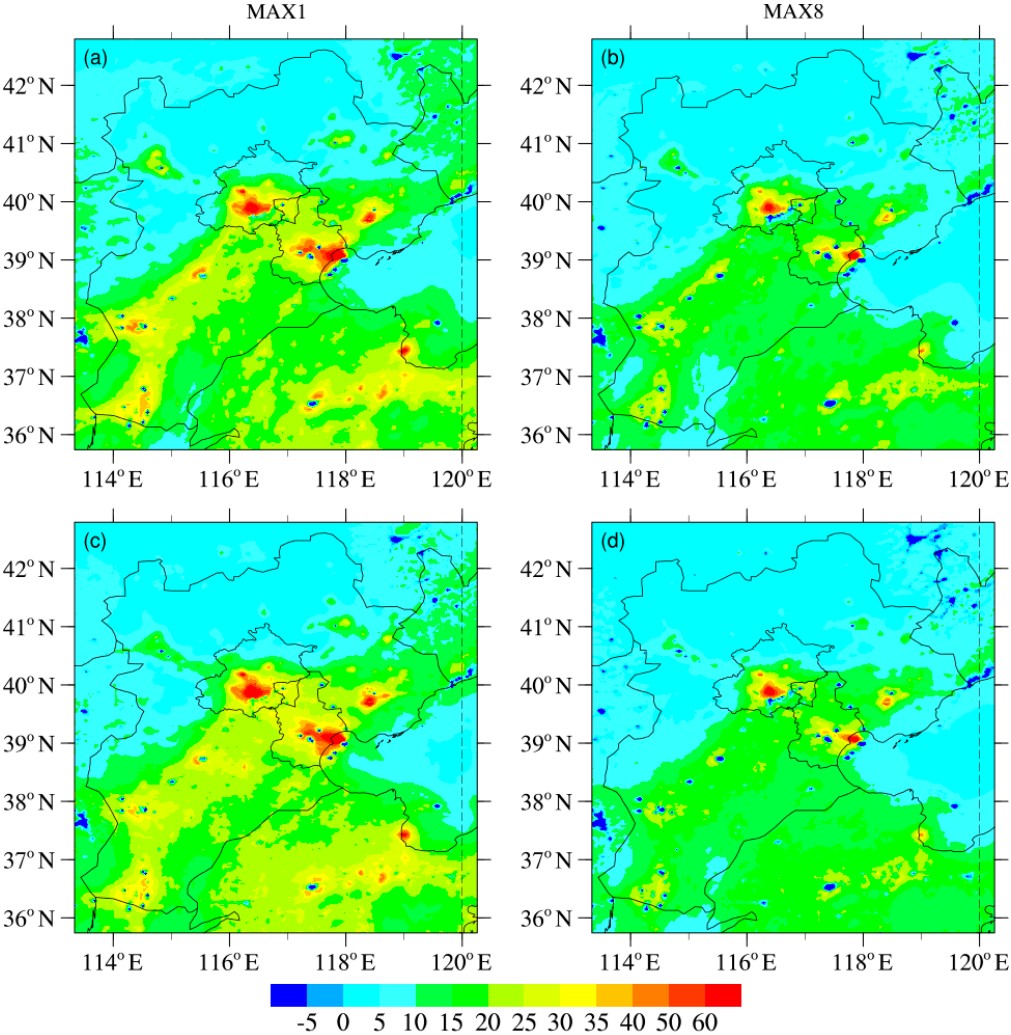

**Figure 12.** Monthly average distribution of photochemical net $O_3$ production (in μg/m$^3$) in June and August 2017 over the NCP; (**a**,**b**) represent the maximum 1 h and maximum 8 h net generation in July; (**c**,**d**) represent the maximum 1 h and maximum 8 h net production in August.

### 3.4. Contributions of Biological Sources

To investigate the impact of BVOCs on ozone formation over the NCP, two simulations were conducted using identical model configurations except that the first scenario considered both anthropogenic and BVOC emissions, while the second included only anthropogenic emissions. Figure 13 illustrates the spatial distribution of the differences in the maximum 1-h average, maximum 8-h average, and monthly average ozone concentrations simulated from the NAQPMS. The largest contributions of BVOCs to the maximum 1-h average, maximum 8-h average, and monthly average were over 30 $\mu g/m^3$ in July and August, suggesting a significant impact of BVOC emissions on ozone formation over the NCP; this result is consistent to a certain degree with the research conducted in the YRD region [44]. In July, the hotspot of BVOC contributions to the maximum 1-h average and the maximum 8-h average was mainly concentrated in the southwestern part of Hebei Province, in the range of 36° N–39° N, 113° E–115° E, and the hotspot of BVOC contributions to the monthly average was concentrated in a larger region. The spatial distribution of BVOC contributions to the maximum 1-h average and maximum 8-h average and monthly average in August is similar to that in July but with lower values due to the weaker solar radiation and lower air temperature in August. The hotspot for the monthly average BVOC contributions is only centered on the southwestern corner of Hebei Province. As shown in Figure 14, there is no major spatial distribution discrepancy between the absolute BVOC contributions and the relative BVOC contributions. The largest contributions of BVOCs to the maximum 1-h average, maximum 8-h average, and monthly average were approximately 18% in July and approximately 12% in August.

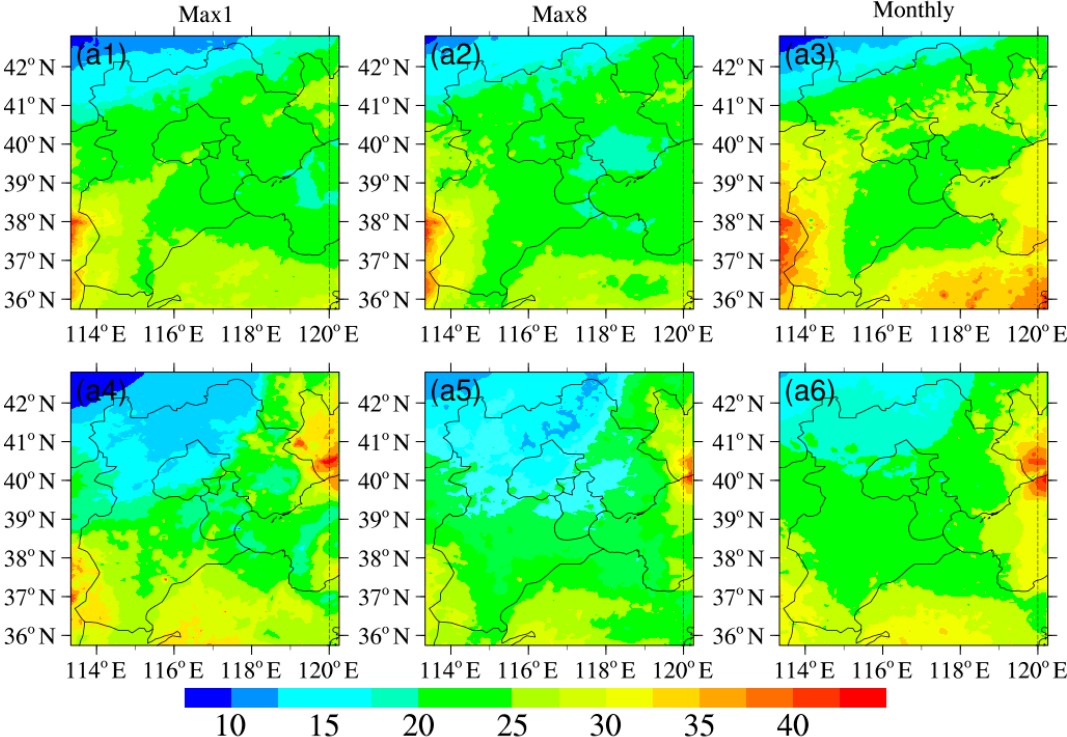

**Figure 13.** Monthly averaged contribution ($\mu g/m^3$) of BVOCs to ozone formation in July and August 2017 over the NCP (**a1**–**a6**); the left column represents the maximum 1-h average, the middle column represents the maximum 8-h average, the right column represents the monthly average, and the rows from top to bottom represent July and August.

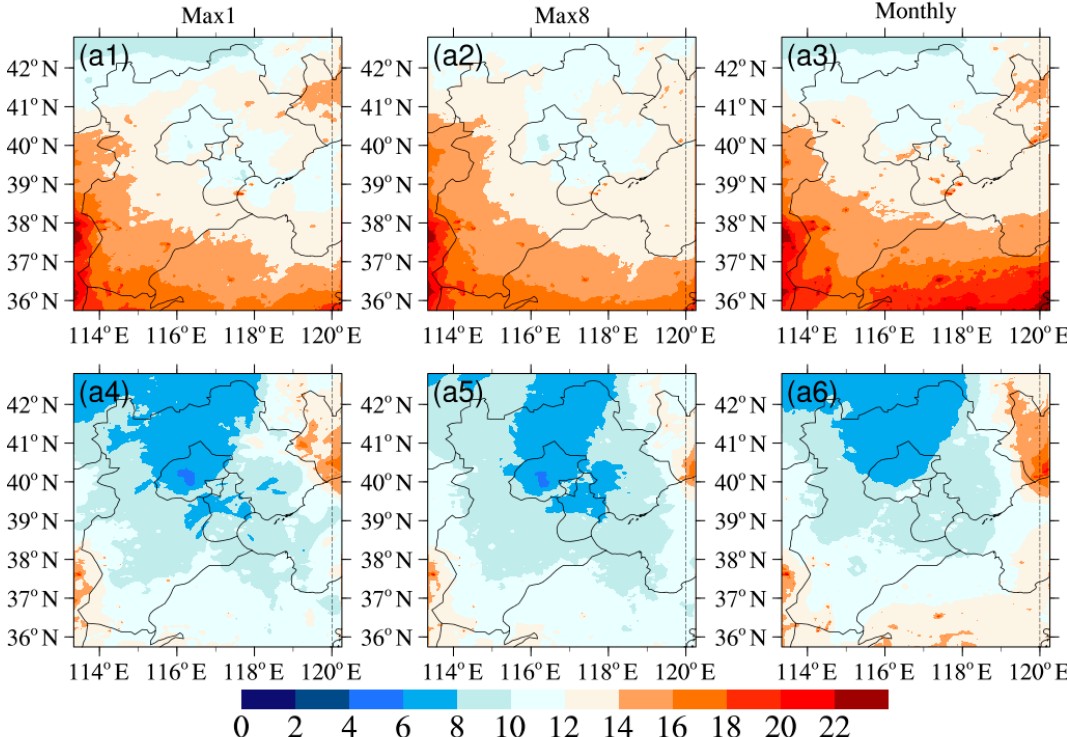

**Figure 14.** Monthly average relative contribution (%) of BVOCs to ozone formation in July and August 2017 over the NCP (**a1–a6**); the left column represents the maximum 1-h average, the middle column represents the maximum 8-h average, the right column represents the monthly average, and the rows from top to bottom represent July and August.

Interestingly, the spatial pattern of BVOC contributions did not align with that of the absolute ozone distribution or with biogenic emissions because ground-level ozone is formed by complex nonlinear reactions between NOx and VOCs. In summer, the amount of BVOC emissions in suburban districts with high vegetation coverage is larger than that in urban districts, and $O_3$ formation in suburban districts and urban districts is commonly controlled by $NO_X$ and VOC. When BVOC emissions in urban districts increase, the increase in $O_3$ levels in urban districts is more pronounced than that in suburban districts, and in some southern regions with large amounts of BVOC emissions, the $O_3$ formation regime will be transformed when BVOC emissions exceed a certain threshold. The discussion above confirms the significance of the $O_3$ formation regime, and we will discuss this issue in detail in Section 4. In addition, it is worth mentioning that the addition of the BVOC emission inventory effectively improves the performance of the simulation shown in Figures 2–4.

## 4. Conclusions

The formation of $O_3$ is sophisticated by nonlinear chemistry that involves multiple ambient pollutants. The measures for $O_3$ pollution prevention should be refined, and the core issue is to determine the main sources and control factors. In this study, we focus on the NCP region, and the Nested Air Quality Prediction Modeling System (NAQPMS) coupled with an online source-tagged model was applied to quantify the $O_3$ source-receptor relationship, to identify the ozone sensitivity threshold with the sensitivity indicator $P(H_2O_2)/P(HNO_3)$ and to investigate the contribution of biogenic emissions to ozone formation. The following conclusions can be drawn:

In general, the simulated $O_3$ was higher during the daytime and in July than at night and in August; the highest concentration, 190 μg/m$^3$, was distributed across the majority of Beijing in July; and the maximum values extended to the region over Chengde and Baoding in Hebei Province.



The results of the source apportionment indicated that the contributions of local emissions and transport from the NCP accounted for the largest proportion of $O_3$, with magnitudes of 25% and 39%, respectively. Compared with the monthly averaged results, the local contribution and regional transport within the NCP during $O_3$ episodes increased by 7% and 10%, respectively. Considering the diurnal variation in the source apportionment, we found that the local contribution was the highest in the daytime, while at nighttime, the contribution of regional transport increased.

Based on the sensitivity tests, two thresholds, 0.08 and 0.2, were detected. The urban sites of Beijing, Tianjin, and the southern part of Hebei Province were controlled by VOCs, with a $P(H_2O_2)/P(HNO_3)$ ratio below 0.08, while the other sites were mainly controlled by $NO_X$, with a $P(H_2O_2)/P(HNO3)$ ratio above 0.2. Similar to the distribution of anthropogenic emissions, the maximum ozone net chemical production was simulated to be approximately 50 $\mu g/m^3$ over the urban sites in Beijing, the eastern part of Tianjin, and the southwestern part of Tangshan, revealing the significant effect of anthropogenic emissions on the net chemical production of ozone.

Biological sources exert an important impact on $O_3$ formation. The maximum absolute contributions for the maximum 1-h average, maximum 8-h average, and monthly average were over 30 $\mu g/m^3$, and a hotspot was centered on southwestern Hebei Province.

In this study, we applied the NAQPMS to analyze the main sources and control factors for $O_3$, but further improvement is needed to compensate for the deficiencies in this work. The uncertainty of the emission inventory affects the simulation performance of the NAQPMS and further affects the sensitivity test results. To obtain a more scientific sensitivity threshold, it is necessary to compare the model results with the observed data combined with a box model and to conduct more sensitivity tests. Third, the ambient aerosol concentration has changed considerably recently, and the effects of radiation and heterogeneous chemistry related to aerosols on $O_3$ formation should be considered in future work.

**Author Contributions:** Conceptualization, Y.Z. (Yuncheng Zhao) and J.L.; methodology, Y.Z. (Yujing Zhang), Y.Z. (Yuncheng Zhao), J.L., Q.W., H.W., H.D., W.Y., Z.W. and L.Z.; formal analysis, Y.Z. (Yujing Zhang), Y.Z. (Yuncheng Zhao) and J.L.; writing-original draft preparation, Y.Z. (Yujing Zhang); writing-review and editing, Y.Z. (Yujing Zhang), Y.Z. (Yuncheng Zhao) and J.L.; visualization, Y.Z. (Yujing Zhang); project administration, J.L.; funding acquisition, J.L. All authors have read and agreed to the published version of the manuscript.

**Funding:** This Work is founded by the China National Key R&D Program (Grant No. 2017YFC0212402 and 2018YFC0213205) and the National Nature Science Foundation of China (Grant No. 91744203, 41620104008).

**Acknowledgments:** We thank anonymous reviewers for their constructive suggestions that helped improve the manuscript.

**Conflicts of Interest:** The authors declare no conflict of interest.

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
