# Peer review of "Modeling Ozone Source Apportionment and Performing Sensitivity Analysis in Summer on the North China Plain"

_atmosphere, doi:10.3390/atmos11090992_

Round 1
Reviewer 1 Report
Review of Modelling Ozone Source Apportionment and Performing Sensitivity Analysis in summer in the North Plain China
General:
Very interesting study to characterize ozone formation and potential sources in summer in North Plain China. The analysis made and the approaches adapted are satisfactory, however would need more clarifications and explanation on how the source apportionment was done and more tests for model performance evaluation are also needed prior to publication of this manuscript.
Specific:
Line 139 : Please specify what are the individual species that represent AVOCs
Line 162 : Please add statistical indicators for NMB (Normalized Mean Bias) and NME (Normalized Mean Errors) and compare the results to other modelling studies performed for china in order to have an idea about the model performance
Line 220 : Please clarify how local vs non-local sources of Ozone are quantified and assessed
Line 332 : What about other photochemical indicators such as NOy and HCHO/NOx what are the values in both NOx and VOC limited regimes and please compare the results obtained for PH2O2/PHNO3 to other studies from the literature
Line 336 : How was investigated the net chemical Ozone production, please explain the methodology
Author Response
We thank the reviewer for your careful read and thoughtful comments on our manuscript. We have carefully taken your comments into considerations in preparing our revision, and below marked in red is our response to your comments point by point, or you can see the attachment (the revised manuscript) for more detail. Thanks again for your comments.
Specific:
Line 139 : Please specify what are the individual species that represent AVOCs
Re: The AVOCs considered in this study include the following species: OLEI (Internal olefin carbons (C=C)); OLET (Terminal olefin carbons (C-C)); PAR (Paraffin carbon); TOL (Toluene); XYL (Xylene). Please see line 137 for more detail.
Line 162 : Please add statistical indicators for NMB (Normalized Mean Bias) and NME (Normalized Mean Errors) and compare the results to other modelling studies performed for china in order to have an idea about the model performance
Re: We have added three statistical parameters of NMB, NME and RMSE to further evaluate the model performance and compared the results to other modelling studies. Please see line 159-166 and Fig2-3 for more detail.
Line 220 : Please clarify how local vs non-local sources of Ozone are quantified and assessed
Re: To differentiate the contributions from various ozone production regions to total ozone levels over CEC, we applied a tagged tracer method in the framework of NAQPMS. Compared with the classic sensitivity analysis that turns on and off emissions in targeted ozone production regions, the tagged tracer method provides a different and more efficient measurement of the relative importance of various ozone production regions and lacks the errors introduced by important non-linearities in the transport and fast photochemistry of ozone and its precursors. Details about the tagged tracer module have been well presented by Li et al.[1]. Please see line 147-153 for more detail.
Line 332 : What about other photochemical indicators such as NOy and HCHO/NOx what are the values in both NOx and VOC limited regimes and please compare the results obtained for PH2O2/PHNO3 to other studies from the literature
Re: We have reviewed other photochemical indicators (NOy, HCHO/NO2) and compared to our results. The transition point between NOx- and VOC-sensitive locations for the NCP occurred at approximately 7.87 ppb NOy, which is close to that for the Barcelona area [2] and San Joaquin Valley [3]. However, the NOy transition value in our study is higher than the result conducted in Lake Michigan, which is considered a scenario with little or no biogenic VOCs[4]. For the ratio of HCHO/NO2, Tonnesen and Dennis[5] found that in situ measurements of the ratios of which between 0.8 and 1.8 indicating a “transition” environment where O3 was both sensitive to radicals and NOX, and the two thresholds we calculated are 0.9 and 2.2. Considering the complexity of the real ambient air, we are supposed to note that differences and uncertainties still exist in applying the indicators to determine ozone formation regime and thresholds often vary by time and region. Please see line 316-332 for more detail.Tonnesen and Dennis [5] found that in situ measurements of the CH2O/NO2 ratio could be used to diagnose local photochemical regimes, with ratios <0.8 indicating a radical-limitedenvironment, ratios >1.8 indicating a NOx-limited environment, and ratios between 0.8 and 1.8 indicating a “transition” environment where O3 was equally sensitive to radicals and NOx.
Line 336 : How was investigated the net chemical Ozone production, please explain the methodology
Re: Here, the net chemical production, N(O3), is calculated by the equation, N(O3) = F(O3)-D(O3)={k1[HO2][NO]+k2[RO2][NO]}-{k3[O(1D)][H2O]+k4[OH][O3]+k4[HO2][O3] +k5[O3][olefin]} in NAQM. The NAQPMS models give the net chemical production as the difference of O3 mixing ratio between the calculation steps of chemistry module with a process analysis package. The net chemical production was calculated in each grid and then average was taken for all the selected grids. Please see line 363-368 for more detail.
References:
[1] Li, J., et al., 2008. Near-ground ozone source attributions and outflow in central eastern China during MTX2006. Atmos. Chem. Phys. 8, 7335–7351.
[2] Jime´nez, P.; Baldasano, J.M. Ozone Response to Precursor Controls in Very Complex Terrains: Use of Photochemical Indicators to Assess O3-NOx-VOC Sensitivity in the Northeastern Iberian Peninsula; J. Geophys. Res. 2004, 109, D20309; doi: 10.1029/2004JD004985.
[3] Sillman, S.; Odman, M.T.; Russell, A.G. Comment on “On the Indicator-Based Approach to Assess Ozone Sensitivities and Emissions Features” by Cheng-Hsuan Lu and Julius S. Chang; J. Geophys. Res. 2001, 106, 20941-20944.
[4] Sillman, S. The Use of NOy, H2O2 and HNO3 as Indicators for the O3-NOx-VOC Sensitivity in Urban Locations; J. Geophys. Res. 1995, 100, 14175-14188.
[5] Tonnesen G S; Dennis R L. Analysis of radical propagation efficiency to assess ozone sensitivity to hydrocarbons and NOx: Local indicators of instantaneous odd oxygen production sensitivity. Journal of Geophysical Research Atmospheres, 2000, 105(D7):9213-9225.
Submission Date
19 July 2020
Date of this revision
17 Aug 2020
Reviewer 2 Report
The authors present a three-dimensional chemical transport study for the O3 tropospheric formation over China aiming to correlate with the local sources of major air pollutants.
The manuscript has some major english errors and misleading use of words that the authors should correct. For example:
1- Line 37: “ in the presence of sunlight in the troposphere”.
2- Line 40: “adverse effects… and the climate change”
3- Line 46: “will worsen in the future”
4-Line 53: “high ozone episodes” (concentrations?)
5-Line 54: “heavy O3 episodes”
Line 226: NOx the “x” should be subscript.
Line 402: “complicated” is not a scientific word.
Line 403: “O3 treatment”.
Lines 404-408: sentence should be rephrased.
Line 408: “made:” cannot be used at the end of sentence.
Line 433: “ Third, the ambient”. I did not find the first and second.
Other comments that should also be considered:
- Line 17: “have been sufficiently”. I cannot agree with the sentence. Even for the PM the large number of studies still reveals discrepancies.
- Line 39: “Recents studies”, should be changed since many of the references are from 2000 and one from 1968.
- Line 40: “adverse effects… and the climate change” the sentence is strange since the authors probably would state that O3 can potentiate climate change. Also the reference [3] is too old. Many recent studies show clearly the correlation that the authors want to stress.
- Along the text, the authors did not take into consideration the significant figures, in particular on the percentages. With the errors that the authors stressed in the conclusions, the contributions cannot be at centesimal place (22.55%, for example).
- On page 4, many web site references are given. They should have the “accessed on …” date, since they are very volatile.
- Missing reference for the comment on Line 165.
- The references should be normalized. Some Refs have issues; some refs have doi; some refs have doi with https; Ref. 8 is incomplete; Ref 10 is published by who? Ref 15 has a “14zhicheng7-157”
Scientifically, the computational model seems appropriate to the study, although I would like to see additional details but assume that the model is well documented over the literature. Some general comments:
- In what concerns the vertical profile (line 112) no units are given for the 30 layers considered. This is extremely relevant for the interpretation of the results.
- Has digital terrain model been considered somehow? The authors report emissions and city contributions, discussing the Beijing mountains. If not considered the elevation model, they should state clearly.
- Another major issue that the authors must comment on is the emission inventory employed, since the conclusions arise from the data chosen. Does the set of data bias the results? What is the relevance of up to date of the input set. Do the conclusions maintain if another year is used? From the website the data is based on 2015 (not 2016) products that span over 1999-2015.
- Having in mind that OH and HO2 species are traditionally associated to participate in competitive reactions, what was the temperature dependent rate constant value for reactions R1,R2 and R3, and how the authors calculated the spatial and which spatial and vertical profiles of this species been employed?
- Since the reactions are temperature dependent, have the authors analysed the impact on temperature uncertainty inert of the WRF model? The same applies to the radiation model for the photochemical processes.
Overall, I think that the manuscript is acceptable for publication after it has been corrected, and the scientific issues properly reported.
Author Response
Re: We thank the reviewer for your careful read and thoughtful comments on our manuscript. We have carefully taken your comments into considerations in preparing our revision, and below marked in red is our response to your comments point by point,or you can see the revised manuscript in attachment for more detail. Thanks again for your comments.
- Line 37: “ in the presence of sunlight in the troposphere”.
Re: the sentence “in the presence of sunlight in the troposphere” has been revised to “with the catalysis of sunlight in the troposphere”
- Line 40: “adverse effects… and the climate change”
Re: please see line 39-45 for more detail.
- Line 46: “will worsen in the future”
Re: the sentence “will worsen in the future” has been revised to “will be worsen in the future”.
- Line 53: “high ozone episodes” (concentrations?)
Re: the sentence “high ozone episodes” has been revised to “ozone episodes”.
- Line 54: “heavy O3 episodes”
Re: the sentence “high ozone episodes” has been revised to “ozone episodes”.
Line 226: NOx the “x” should be subscript.
Re: the issue of subscript has been revised.
Line 402: “complicated” is not a scientific word.
Re: the word “complicated” has been replaced by “sophisticated”.
Line 403: “O3 treatment”.
Re: the word “treatment” has been replaced by “pollution prevention”.
Lines 404-408: sentence should be rephrased.
Re: please see line 408-411 for more detail.
Line 408: “made:” cannot be used at the end of sentence.
Re: the word “made” has been replaced by “drawn”.
Line 433: “ Third, the ambient”. I did not find the first and second.
Re: We have revised this error and replaced the “Third” as “The last”.
Other comments that should also be considered:
- Line 17: “have been sufficiently”. I cannot agree with the sentence. Even for the PM the large number of studies still reveals discrepancies.
Re: We agree and accept the reviewer's views, and we changed the sentence by the rigorous statement “In recent years, the issues of haze have long been the most important environmental problems and caused widespread concern.”
- Line 39: “Recent studies”, should be changed since many of the references are from 2000 and one from 1968.
Re: We have updated the old references.
- Line 40: “adverse effects… and the climate change” the sentence is strange since the authors probably would state that O3 can potentiate climate change. Also the reference [3] is too old. Many recent studies show clearly the correlation that the authors want to stress.
Re: we rephrased the involving sentences and updated the old references. Please see line 39-45 for more detail.
- Along the text, the authors did not take into consideration the significant figures, in particular on the percentages. With the errors that the authors stressed in the conclusions, the contributions cannot be at centesimal place (22.55%, for example).
Re: We have checked and corrected the significant figures in the manuscript.
- On page 4, many web site references are given. They should have the “accessed on …” date, since they are very volatile.
Re: We have annotated the visit date of the website. Please see line 130 and line 136 for more detail.
- Missing reference for the comment on Line 165.
Re: We have added the reference for the comment.
- The references should be normalized. Some Refs have issues; some refs have doi; some refs have doi with https; Ref. 8 is incomplete; Ref 10 is published by who? Ref 15 has a “14zhicheng7-157”
Re: We carefully checked and standardized the reference format.
Scientifically, the computational model seems appropriate to the study, although I would like to see additional details but assume that the model is well documented over the literature. Some general comments:
- In what concerns the vertical profile (line 112) no units are given for the 30 layers considered. This is extremely relevant for the interpretation of the results.
Re: Vertically, the model employed terrain-following coordinate, and the vertical grid spacing was divided into 30 layers increasing gradually from 20m near the surface to 20km at the top with 20 layers in the boundary layer. Please see line 116-118 for more detail.
- Has digital terrain model been considered somehow? The authors report emissions and city contributions, discussing the Beijing mountains. If not considered the elevation model, they should state clearly.
Re: The governing model equations for tracer continuity in NAQPMS are formulated in a spherical and terrain-following coordinate [1]. The terrain -following coordinate [2] has advantages in implementing boundary conditions and consider the terrain influence. As shown in the figure, the vertical grid structure consists of 30 layers from the surface to the model top with the lowest 20 layers in the boundary layer to provide a more detailed characterization of terrain effect. Please see line 92-94 for more detail.
- Another major issue that the authors must comment on is the emission inventory employed, since the conclusions arise from the data chosen. Does the set of data bias the results? What is the relevance of up to date of the input set. Do the conclusions maintain if another year is used? From the website the data is based on 2015 (not 2016) products that span over 1999-2015.
(1)The MEIC (Multi-resolution emission inventory for China) was used in this study to investigate the emission responses to ozone concentration. MEIC is a dynamic technology-based inventory developed for China covering the years from 1990 to recent years by Tsinghua University [3] covering 10 major air pollutants and greenhouse gases (SO2, NOx, CO, NMVOC, NH3, CO2, PM2.5, PM10, BC and OC) and more than 700 anthropogenic emission sources. The MEIC has several updates, such as a unit-based emission inventory of power plants [4], a high-resolution vehicle emission inventory at the county level [5] and an improved non-methane volatile organic carbon (NMVOC) speciation approach for various chemical mechanisms [6]. Considering that China has been vigorously controlling pollution in recent years, the MEIC inventory includes recent control policies based on the available official reports to provide a reliable description of pollutants emission. In general, the MEIC is a technologically advanced, reliable and widely used inventory.
(2)Exactly, the latest MEIC has covered the year of 2016, but the data updated in the website is lagging, we fortunately get the data collection by personal contact and applied it to this study.
(3): We cannot deny that the uncertainties in emission inventory are major factors causing simulation discrepancies in recent three-dimensional chemical transport modes. Note that the emission difference between adjacent years will not be very large.
- Having in mind that OH and HO2 species are traditionally associated to participate in competitive reactions, what was the temperature dependent rate constant value for reactions R1,R2 and R3, and how the authors calculated the spatial and which spatial and vertical profiles of this species been employed?
Re: The NAQPMS we used in this study utilized the CBMZ gas-phase chemical reaction scheme which is composed of 133 reactions for 53 species providing a detailed description of tropospheric O3-NOX- hydrocarbon chemistry, and the rate constant for reactions R1, R2, R3 can also be found in the reference [7]. NAQPMS is capable of calculating and outputting the values of OH and HO2 online for each time step and each grid without assumptions.
- Since the reactions are temperature dependent, have the authors analysed the impact on temperature uncertainty inert of the WRF model? The same applies to the radiation model for the photochemical processes.
Re: The NAQPMS is driven by the mesoscale numerical weather prediction system-WRF, and the simulation of meteorological field is vital to the simulation of pollutants, especially to the species sensitive to temperature. Therefore, we evaluate the performance of WRF at 6 stations consisting of 3 rural sites and 3 urban sites with the MICAPS data by calculating several important statistical parameters. We can see that WRF rationally recaptured the magnitudes of the air temperature with R up to 0.9, and the MB and the RMSE of most sites is around 4℃ and 5℃, providing sufficient confidence for the following discussion.
|
Station |
Longitude(°E) |
Latitude(°N) |
R |
MO |
MP(℃) |
MB(℃) |
RMSE |
NMB(%) |
|
Beijing |
116.28 |
39.93 |
0.9 |
26.6 |
31.4 |
4.9 |
5.5 |
18.3 |
|
Chaoyang |
116.48 |
39.95 |
0.9 |
27.1 |
31.2 |
4.1 |
4.2 |
15.3 |
|
Haidian |
116.28 |
39.98 |
0.9 |
26.0 |
31.5 |
5.5 |
4.7 |
21.0 |
|
Huairou |
116.63 |
40.32 |
0.9 |
25.5 |
30.3 |
4.7 |
5.7 |
18.5 |
|
Changping |
116.22 |
40.22 |
0.9 |
26.3 |
31.0 |
4.6 |
5.4 |
17.6 |
|
Shijingshan |
116.18 |
39.93 |
0.9 |
26.1 |
31.3 |
5.8 |
4.7 |
20.0 |
Since the ozone formation is temperature dependent, we also simply analysed the impact of temperature uncertainty inert of the WRF model on the ozone simulation. The averaged observed ozone generally increased with the increasing air temperature, and the averaged simulated and observed air temperature are 31℃and 26℃, respectively. The discrepancy caused by uncertainty in air temperature of ozone is about 30μg/m3, which is within an acceptable range. Please see the attachment for more detail.
References:
[1] Wang, Z., Maeda, T., Hayashi, M., Hsiao, L.F., & Liu, K.Y., 2001. A nested air quality prediction modeling system for urban and regional scales, application for high-ozone episode in Taiwan. Water Air Soil Pollut. 130, 391–396.
[2] Phillips, N. A. A COORDINATE SYSTEM HAVING SOME SPECIAL ADVANTAGES FOR NUMERICAL FORECASTING[J]. Journal of Atmospheric ences, 1957, 14(2):184-185, doi: 10.1175/1520-0469(1957)014<0184:ACSHSS>2.0.CO;2.
[3] Zhang, Q., Streets, D. G., Carmichael, G. R., He, K. B., Huo, H., Kannari, A., Klimont, Z., Park, I. S., Reddy, S., Fu, J. S., Chen, D., Duan, L., Lei, Y., Wang, L. T., and Yao, Z. L.: Asian emissions in 2006 for the NASA INTEX-B mission, Atmos. Chem.Phys., 9, 5131–5153, doi:10.5194/acp-9-5131-2009, 2009.
[4] Liu, F., Zhang, Q., Tong, D., Zheng, B., Li, M., Huo, H., and He, K. B.: High-resolution inventory of technologies, activities, and emissions of coal-fired power plants in China from 1990 to 2010, Atmos. Chem. Phys., 15, 13299–13317, doi:10.5194/acp-15-13299-2015, 2015.
[5] Zheng, B., Huo, H., Zhang, Q., Yao, Z. L., Wang, X. T., Yang, X. F., Liu, H., and He, K. B.: High-resolution mapping of vehicle emissions in China in 2008, Atmos. Chem. Phys., 14, 9787–9805, doi:10.5194/acp-14-9787-2014, 2014.
[6] Li, M., Zhang, Q., Streets, D. G., He, K. B., Cheng, Y. F., Emmons, L. K., Huo, H., Kang, S. C., Lu, Z., Shao, M., Su, H., Yu, X., and Zhang, Y.: Mapping Asian anthropogenic emissions of nonmethane volatile organic compounds to multiple chemical mechanisms, Atmos. Chem. Phys., 14, 5617–5638, doi:10.5194/acp-14-5617-2014, 2014.
[7] Zaveri, R.A., & Peters, L.K., 1999. A new lumped structure photochemical mechanism for large-scale applications. J. Geophys. Res. 104, 30,387–30,415.
Submission Date
19 July 2020
Date of this revision
17 Aug 2020
Round 2
Reviewer 2 Report
The manuscript has been improved by the authors who attempt to make a better presentation of the work and provided additional correlations. Since scientifically sounds accurate I think that is now accepted for publication.